# Time encoding migrates from prefrontal cortex to dorsal striatum during learning of a self-timed response duration task

Gabriela C Tunes[1†], Eliezyer Fermino de Oliveira[1†], Estevão UP Vieira[1], Marcelo S Caetano[1,2], André M Cravo[1], Marcelo Bussotti Reyes[1]*

[1]Center for Mathematics, Computing, and Cognition, Universidade Federal do ABC, Sao Bernardo do Campo, Brazil; [2]Instituto Nacional de Ciência e Tecnologia sobre Comportamento, Cognição e Ensino, Brazil

**Abstract** Although time is a fundamental dimension of life, we do not know how brain areas cooperate to keep track and process time intervals. Notably, analyses of neural activity during learning are rare, mainly because timing tasks usually require training over many days. We investigated how the time encoding evolves when animals learn to time a 1.5 s interval. We designed a novel training protocol where rats go from naive- to proficient-level timing performance within a single session, allowing us to investigate neuronal activity from very early learning stages. We used pharmacological experiments and machine-learning algorithms to evaluate the level of time encoding in the medial prefrontal cortex and the dorsal striatum. Our results show a double dissociation between the medial prefrontal cortex and the dorsal striatum during temporal learning, where the former commits to early learning stages while the latter engages as animals become proficient in the task.

*For correspondence:
marcelo.reyes@ufabc.edu.br

†These authors contributed equally to this work

## Editor's evaluation

This study investigates the question of whether distinct brain areas differentially encode time during the learning of a simple motor timing task. The key novel result is that early in training the dynamics of the medial prefrontal cortex provides the best code for time, but later in training the striatum provides a better code. In addition, the article reports that the inactivation of medial prefrontal cortex produces a delayed learning effect, while the inactivation of the striatum after learning led to impairment of performance. Thus, the observation that temporal coding and the necessity of brain area for task performance transfers from medial prefrontal cortex to the striatum during learning is an important observation for the field.

## Introduction

Even though keeping track of time is essential for survival, our understanding of how animals encode temporal information in terms of neuronal activity is still modest (*Ivry and Spencer, 2004*; *Merchant et al., 2013a*). The literature contains evidence of physiological activity associated with timing, as, for example, ramping neurons (*Narayanan and Laubach, 2009*; *Kim et al., 2013*; *Emmons et al., 2017*) and time cells (*MacDonald et al., 2011*; *Eichenbaum, 2014*), neuronal oscillations (*Matell et al., 2003*), and sequential firing of neurons (*Modi et al., 2014*; *Zhou et al., 2020*). Most of these encoding patterns are particular cases of a general type of encoding called population clocks, which states that any reliable dynamical evolution of neural activity works as a potential clock and might serve as a

timing mechanism (*Buonomano and Maass, 2009*). However, establishing causal links between the time encoding and timing itself has been elusive (*Monteiro, 2020*; *Modi et al., 2014*).

Reports of the spiking activity of neurons involved in keeping track of time implicate regions like frontal cortex (*Emmons et al., 2017*; *Kim et al., 2017*; *Matell et al., 2003*; *Brody et al., 2003*; *Xu et al., 2014*; *Wang et al., 2018*; *Kunimatsu and Tanaka, 2012*), motor cortex (*Lebedev et al., 2008*; *Laubach et al., 2000*; *Zhou et al., 2020*; *Merchant et al., 2013b*), striatum (STR) (*Mello et al., 2015*; *Matell et al., 2003*; *Bakhurin et al., 2017*; *Gouvêa et al., 2015*; *Emmons et al., 2017*; *Zhou et al., 2020*; *Merchant et al., 2013b*), hippocampus (*MacDonald et al., 2011*; *Eichenbaum, 2014*), thalamus (*Komura et al., 2001*; *Tanaka, 2007*), substantia nigra pars compacta (*Soares et al., 2016*), among others. Moreover, studies with humans showed that distinct tasks of time estimation can activate different parts of the brain (*Merchant et al., 2008*). Active manipulations of the physiological activity have also helped assess the involvement of these areas in timing, particularly in the medial prefrontal cortex (mPFC) (*Buhusi et al., 2018*; *Soares et al., 2016*; *Kim et al., 2009*; *Emmons et al., 2017*) and STR (*Gouvêa et al., 2015*). However, because timing tasks usually require many training sessions, the neurophysiological process underlying their acquisition has been less studied. The vast majority of the electrophysiological recordings in timing tasks come from well-trained animals (but see *Modi et al., 2014*). However, observing how neuronal activity develops with learning can considerably improve our understanding of brain mechanisms underlying timing.

We investigated how spiking activity in the mPFC and STR encodes a time interval during learning. We trained rats to sustain responses for at least 1.5 s and investigated how well the two brain areas decode the time interval during acquisition. If a brain area is involved in the encoding of that interval, a trial-dependent and more structured spiking activity must emerge due to learning. As a consequence, the performance of algorithms that evaluate the decoding should increase. Contrary to evidence of the concomitant involvement of mPFC (*Buhusi et al., 2018*; *Kim et al., 2009*) and STR (*Mello et al., 2015*) in timing tasks, our results show very different roles of these two regions during learning. As training progresses, the decoding performance based on electrophysiological recordings decreases in the mPFC while it increases in the STR. Such results were confirmed and further investigated with pharmacological experiments. Our studies provide a useful method to investigate electrophysiological correlates of temporal learning and advance our understanding of the role of mPFC and STR in interval timing.

## Results

### Rats learn to time in a single session

We conceived a novel experimental design in which rats improve their timing in a single session (*Reyes et al., 2020*), allowing us to track the activity of individual neurons during learning (*Figure 1*). Animals had to remain in a nose poke for at least 1.5 s to receive access to the sucrose solution, limited to three licks, after which the access gate closes (*Figure 1A*). Shorter responses produced no consequences. Critically, we recorded activity before, during, and after animals had learned the critical interval.

We implanted two sets of animals, one with electrode arrays in the mPFC (group mPFC, N = 4 animals) in long sessions (> 4 hours) and another with arrays both in the mPFC and STR (group mPFC + STR, N = 4 animals) in shorter sessions (131 neurons in total). Animals produced between 801 and 1671 trials (878 trials on average) in long sessions (group mPFC), and, in shorter sessions (group mPFC + STR), between 436 and 936 (606 on average) on day 1 and between 381 and 699 (535 on average) trials on day 2.

All eight animals showed significant learning in the first session, producing longer lever presses $T$ and yielding higher reward rates late in the session ($t(7) = 7.9$, $p = 10^{-4}$, Cohen's d = 3.2, $\overline{T}_{\text{early}} = 0.99 \pm 0.06$ s, $\overline{T}_{\text{late}} = 1.71 \pm 0.08$ s; *Figure 1B and C*).

Animals trained for 2 days (group mPFC + STR, N = 4) also produced longer responses late in the first session as compared to the beginning (*Figure 1C*, group mPFC + STR, session 1). In the second session, these rats' responses were initially shorter than those at the end of session 1, increasing throughout the session. Hence, animals retained information between sessions 1 and 2. Furthermore, rats performed similarly at the end of both sessions, suggesting that animals learned the task almost to their best on the first day of training. A repeated-measures ANOVA showed an effect of stage ($F_{[1,3]} = 61.0$, $p = 0.004$, $\eta_p^2 = 0.95$), no effect for day ($F_{[1,3]} = 2.37$, $p = 0.22$, $\eta_p^2 = 0.44$), and an

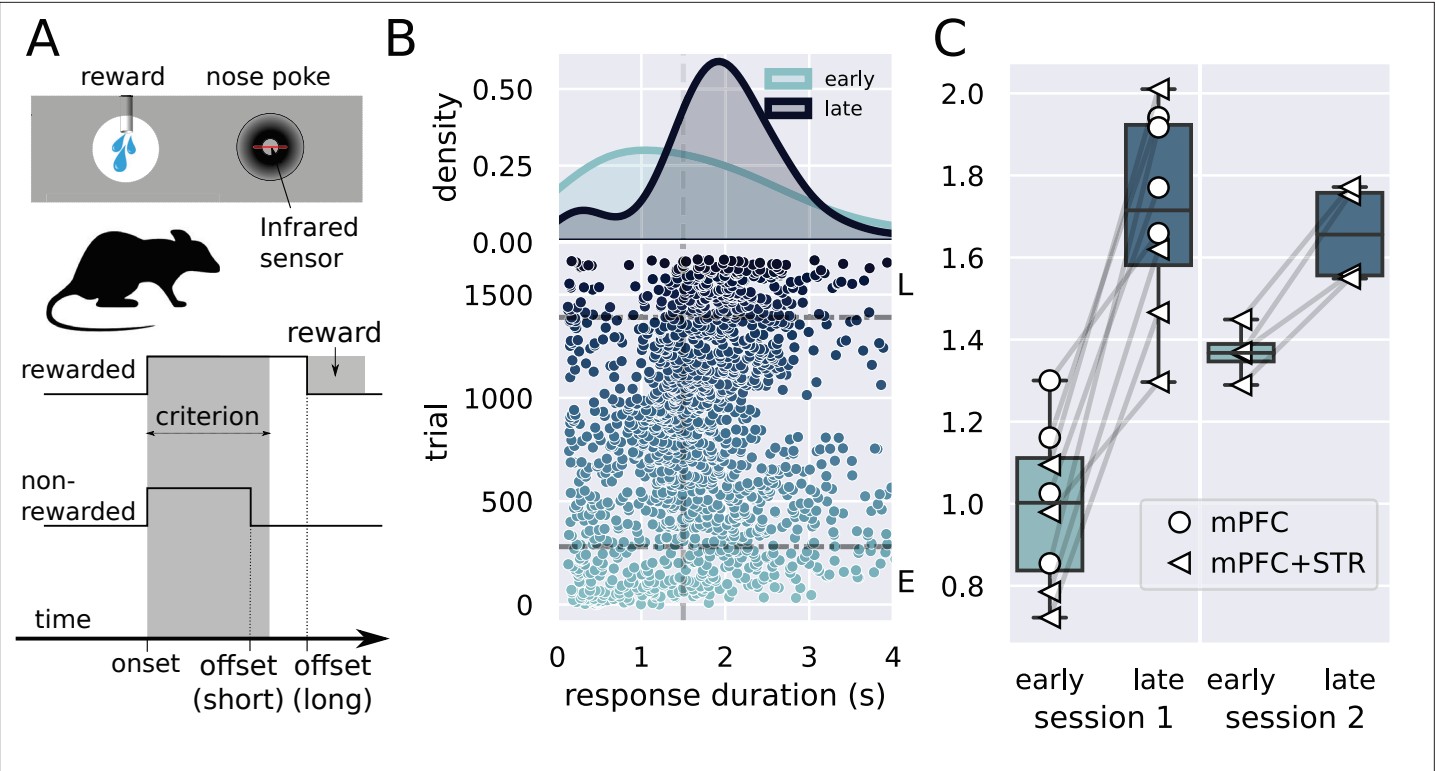

**Figure 1.** Description of the task and behavioral results. (**A**) Experimental design used in the experiment. The trial starts when the rat inserts its snout in the nose poke installed next to the reward port. The step lines represent the nose poke onset and offset. Depending on how long the rat stays in the nose poke—the response duration—it may receive access to a sucrose solution. Responses lasting more than 1.5 s grant access to three licks of sucrose solution, while shorter responses produce no consequences. (**B**) Responses from a single rat in the first session of training in the task. (Lower graph) Each dot represents a response duration (x-axis) in a specific trial (y-axis). The vertical dashed line represents the criterion (1.5 s) for receiving the reward. The two horizontal dash-dotted lines show the trials that were used in the electrophysiology experiments for early 'E' and late 'L.' (Upper graph) Probability density of responses as a function of the duration for the early and late trials, showing that the distribution shifted to the right. (**C**) Mean response duration for individual rats early and late in sessions 1 and 2. All animals significantly increased their responses in the very first session of training. For one group of animals (group medial prefrontal cortex [mPFC] + striatum [STR]), we recorded a second day of training. At the beginning of this second session (early trials), animals' response durations were shorter than the late trials of session 1, but much longer than the early trials, showing that the training from session 1 improved the behavior early in session 2. Throughout the second session, rats achieved the same performance as the late trials of session 1, suggesting that the training in the first session took the rats close to the optimal performance.

interaction stage versus day ($F_{[1,3]} = 26.0$, $p = 0.014$, $\eta_p^2 = 0.89$). A post-hoc analysis revealed a significant difference between early and late responses in the first ($t(3) = 7.02$, $p = 0.006$), but only marginally significant in the second session ($t(3) = 2.90$, $p = 0.06$).

## Time representation decreases with learning in the medial prefrontal cortex, but not in the striatum, during the first session

We investigated the progression of individual neurons' activity in the mPFC and the dorsal striatum during learning (*Figure 2*). We examined how the neuronal activity in both regions were modulated during trials and how such modulation evolved with learning (*Figure 2C–F* for mPFC and *Figure 2G–J* for STR).

Neurons exhibited diverse activity patterns, with a small number of units modulated by the onset or offset of the nose poke and others that became less responsive to trials during learning. Notably, in the mPFC, some neurons' activity climbed up or down during the initial trials but became flat late in training. Overall, there was no evidence that the climbing activity increased with training, neither in the mPFC ($t(3) = 0.54$, $p = 0.60$, $\overline{N}_{early} = 13 \pm 8$ s, $\overline{N}_{late} = 9.5 \pm 7.4$ s) nor in the STR ($t(3) = -0.081$, $p = 0.94$, $\overline{N}_{early} = 4.5 \pm 3.8$ s, $\overline{N}_{late} = 4.75 \pm 3.6$ s). These results were at odds with our working hypotheses that the mPFC and the STR were highly involved in keeping track of time and that, once learning

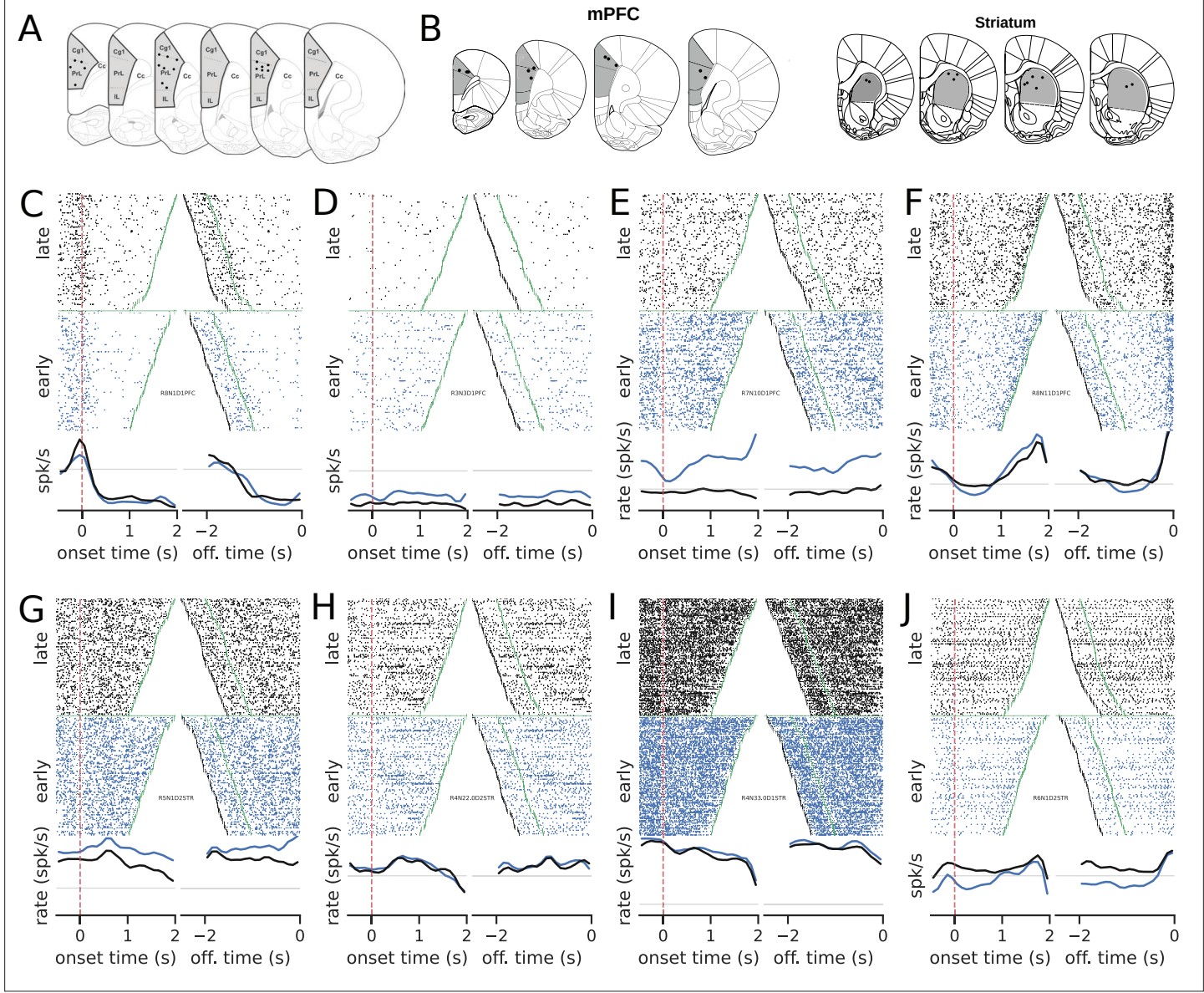

**Figure 2.** Recording sites and examples of neural activity during trials. (**A**) Recording sites in the medial prefrontal cortex (group mPFC). (**B**) Recording sites for group mPFC + striatum (STR). (**C–J**) The upper graphs show the raster plots from early (blue) and late (black) trials for mPFC (**C–F**) and STR (**G–J**). The bottom graphs show the corresponding firing rates, color-coded. The time scale for raster plots and firing rates consists of recordings starting 500 ms before the rat enters the nose poke and ending when the animal leaves the nose poke. Trials were either aligned to the nose poke entering (onset time, left) or to the instant the rat left the nose poke (offset time, right). In onset time, the zero represents the instant the rat entered and the green points show when the animal left the nose poke. In offset time, the zero represents the instant the rat left the nose poke and, consequently, all times are negative. The green point shows when the animal entered the nose poke and the black markers at the beginning of each trial represent the 500 ms before entering the nose poke. (**C**) A neuron whose activity is tuned to the trial onset. (**D–F**) Examples of neurons whose activity within the trial became less sensitive (flatter) in later trials compared with earlier trials in the session.

took place, an encoding scheme should emerge. However, the evidence for mPFC data pointed to a very distinct scenario: a vanishing neural code.

Given the heterogeneity of individual neurons' responses and based on recent proposals of population clocks, we explored whether the pattern of activity in the mPFC and STR could work as a stable representation of time (***Figure 3***). We used spikes from all recorded neurons (***Table 1***) during reinforced trials, that is, trials longer than the criterion (1.5 s) and shorter than 3.5 s (a similar anlaysis made with non-reinforced trials is shown in supplementary material). We then truncated all trials at 1.5 s and calculated the firing rate in bins of 100 ms. This procedure results in a matrix of firing rates,

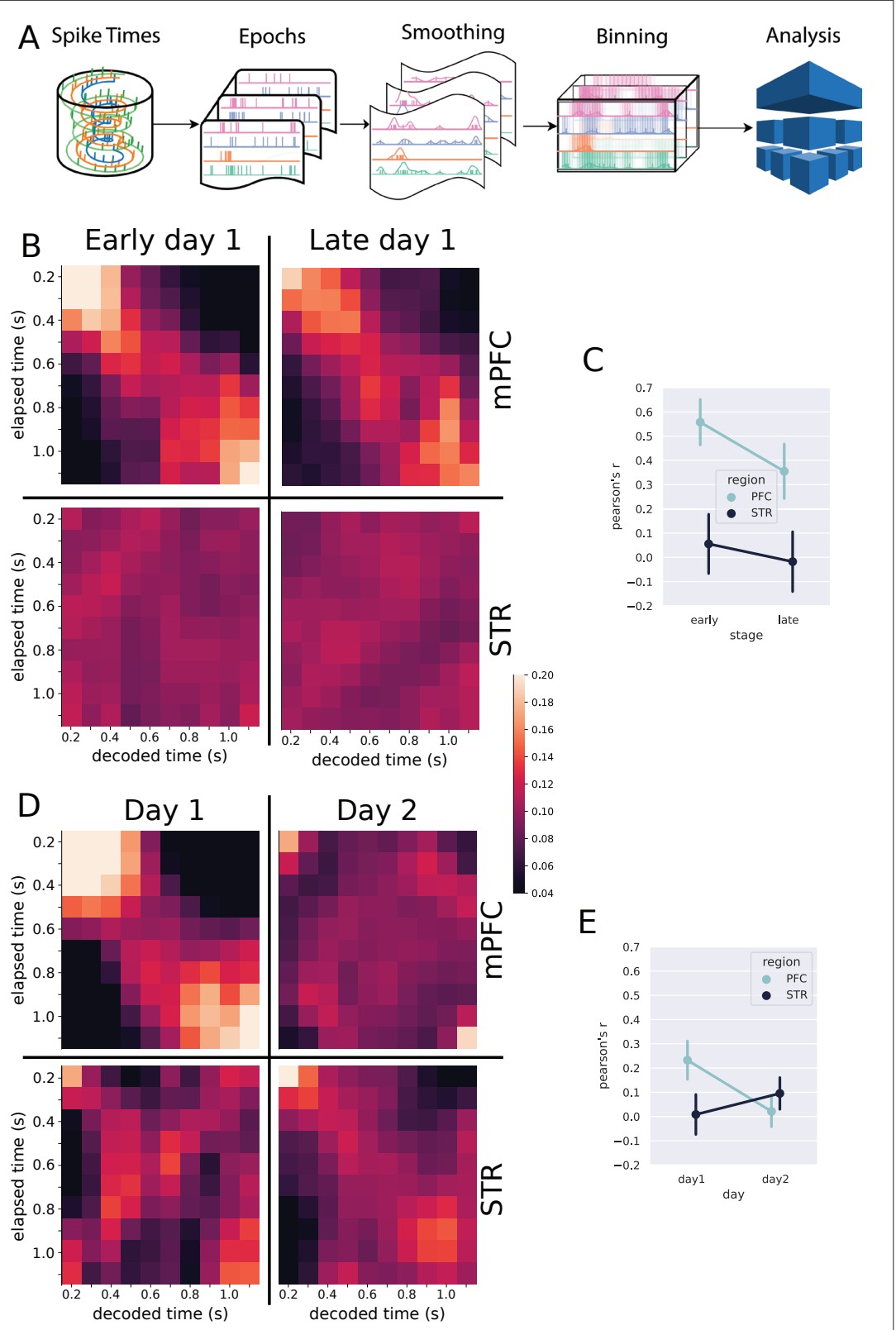

**Figure 3.** Classification analysis. (**A**) Pipeline of the analysis at the population level. Spike trains are epoched according to the trials, from 0.5 s before nose poke onset through the offset. The epoched trials are shown in *Figure 2C–J*. The trials are smoothed with a Gaussian kernel (σ = 100 ms) and then binned every 100 ms. The resulting spike rates were analyzed with the linear discriminant analysis. (**B, D**) Results of the classification decoder

*Figure 3 continued on next page*

*Figure 3 continued*

analysis within session 1 (**B**) and between sessions 1 and 2 (**D**). The confusion matrix (in the form of a heatmap) shows the probability of decoding a particular actual time bin (x-axis) as one of the possible decoding bins (y-axis). A diagonal pattern of bright pixels indicates a better performance of the classifier. (**B**) Results of the classifier in the medial prefrontal cortex (mPFC) (top row) and striatum (STR) (bottom row), early and late in training showing that the performance decreased with learning. (**C**) Overall classifier performance for mPFC (blue line) and STR data (black line), measured as a correlation coefficient (Pearson's R) between the actual and the predicted bin. The results show that the performance decreases in the mPFC while there is no evidence of changes in the STR. Error bars represent the standard deviation of the distributions obtained with a bootstrap procedure (1000 folds). (**D, E**) Same analysis as in (**B, C**) but comparing data from sessions 1 and 2, for animals with electrodes implanted in both regions (group mPFC + STR). (**E**) Decoding performance from mPFC (top row) and STR (bottom row) during day 1 (left column) and day 2 (right column). During day 1, mPFC outperformed the STR in decoding time information. On day 2, the opposite pattern emerged. (**E**) Pearson's R score of the decoder performance summarizing results shown in (**E**).

representing the discrete bins, for all neurons, in all trials. We used these data to train a machine-learning model (linear discriminant analysis [LDA]) to decode the time bin using firing rates as inputs and measured the classifier's performance as the correlation coefficient between the predicted bin and the actual time bin. We calculated Pearson's R using 1000 different training and testing sets, generating a distribution of R values compared between groups and conditions.

The classifier reliably decoded the elapsed time from mPFC neurons (*Figure 3B, C*), both early and late in session 1 (Pearson's $R_{early} = 0.56$, ci (mean) = $[0.37, 0.74]$; $R_{late} = 0.35$, ci (mean)= $[0.14, 0.56]$), suggesting that this region encoded time from very early stages of training. These results were obtained by using all neurons from all rats. We analyzed the robustness of the results between animals using a jackknife procedure, which showed a marginal evidence that the decoding performance decayed during the first day of training ($t(7) = 2.313, p = 0.054$).

The classifier revealed a distinct scenario for the STR in the first session. We did not find evidence that striatal neurons were decoding the time intervals (Pearson's $R_{early} = 0.05$, ci (mean)= $[-0.19, 0.29]$, $R_{late} = 0.02$, ci (mean)= $[-0.27, 0.22]$) (*Figure 3B*, lower row). The jackknife analysis did not show significant differences between stages in the first session ($t(3) = 0.49, p = 0.64$). These results suggest that the prefrontal cortex exceeded the STR in time decoding in session 1. Such results, however, should be taken with caution because the number of neurons recorded in the mPFC was greater than that in the STR, a factor that affects the encoding performance.

Beyond the classifier's absolute predicting power, affected by the number of neurons recorded, we were interested in how the performance changed in each region as the animals learned on the first day of training—going from a naive animal to almost a proficient level. Since we record from the same neurons in each animal, a better performance in late trials would reveal that the region developed a time-related neural code. Our data provide some evidence that the mPFC and STR evolved differently in the session. The performance of the classifier in the mPFC decreased with training (*Figure 3C*), reinforcing the previous evidence that neurons in the mPFC disengaged from timing as the training progressed. On the other hand, we could not find evidence that the STR engaged the timing task during the first day of learning, neither early nor late in the session.

We further analyzed the decoder performance in the two brain regions during two consecutive training days (*Figure 3D and E*). For these recordings, we only used rats from group mPFC + STR, for which all rats had simultaneous recordings in both brain regions ($N = 4$).

The performance decreased in the mPFC from the first to the second session, passing from significant performance ($R_{day1} = 0.22$, ci (mean) =

**Table 1.** Number of units selected for analysis per rat, brain region, and day of training.

Rats 7–10 received only one session of training, with recordings in medial prefrontal cortex (PFC). Rats 3–6 trained for two days, with recordings in PFC and striatum (STR). The symbol '-' denotes that no neurons in that specific condition were recorded.

| | Rat | 3 | 4 | 5 | 6 | 7 | 8 | 9 | 10 |
|---|---|---|---|---|---|---|---|---|---|
| Day | Region | | | | | | | | |
| 1 | PFC | 4 | 22 | 5 | 4 | 11 | 35 | 10 | 9 |
| | STR | 12 | 15 | 16 | 1 | - | - | - | - |
| 2 | PFC | 14 | 28 | 5 | 3 | - | - | - | - |
| | STR | 16 | 16 | 23 | 5 | - | - | - | - |
| Total | | 46 | 81 | 49 | 13 | 11 | 35 | 10 | 9 |

$[0.06, 0.39]$) to null ($R_{\text{day2}} = 0.02$, ci (mean) = $[-0.11, 0.15]$). However, this difference was not significant when the variability between rats was considered ($t(3) = 0.784$, $p = 0.46$). The decoding performance in the STR neurons—which was almost negligible in the first session—slightly increased by the second session ($R_{\text{day1}} = 0.01$, ci (mean) = $[-0.16, 0.17]$; $R_{\text{day2}} = 0.09$, ci (mean) = $[-0.03, 0.22]$). When considering the variability between animals, there was no significant difference between days ($t(7) = -0.852$, $p = 0.422$).

## Medial prefrontal cortex is crucial for the acquisition but not the expression of timed responses

Given the unexpected finding of how training modulated activity in the mPFC, we further investigated its role in learning, looking for causal evidence supporting the results obtained with electrophysiological recordings. We designed a 5-day training protocol to assess the temporal performance in the first days of training with the mPFC under muscimol inactivation. We used the behavioral task shown in *Figure 1A*, except that the rats responded to a lever instead of a nose poke, and received a sugar pellet as reward. In the three first sessions, the experimental group (muscimol group, N=8) received muscimol (100 ng/0.5 μl) infusion in the mPFC, while the control animals (saline group, N=11) received saline (0.5 μl). On the fourth day, both groups received saline, and in the fifth session, both received muscimol.

The mPFC inactivation severely impaired learning, but had no effect on the performance of trained animals (*Figure 4*). Although both saline and muscimol rats produced short responses early in training, only rats in the saline group made longer responses as the session progressed (*Figure 4B*). In the first session, a mixed ANOVA showed an effect for group ($p = 0.048$, $F_{[1,17]} = 4.5$, $\eta_p^2 = 0.21$), stage ($p = 0.0006$, $F_{[1,17]} = 17.5$, $\eta_p^2 = 0.51$), and an interaction between group and stage ($p = 0.0049$, $F_{[1,17]} = 10.5$, $\eta_p^2 = 0.38$). A Holm's post-hoc correction showed that the groups differed late ($p = 0.023$) but not early in the session ($p = 0.91$). Also, the muscimol group displayed no significant difference in the response duration comparing early versus late trials ($t(7) = 0.36$, $p = 0.73$, paired $t$-test), with an anecdotal Bayes factor favoring the null hypothesis ($BF_{10} = 0.36$). Therefore, the saline group learned in the first session, while the muscimol group displayed no sign of change in behavior.

The effect of mPFC inactivation persisted over the first three sessions, impairing learning (*Figure 4C*). Even though both groups learned over the three sessions, shown by an effect of session ($F_{[2,34]} = 20.8$, $p = 10^{-6}$, $\eta^2 = 0.55$, mixed ANOVA), rats from the saline group produced longer responses compared with the muscimol group (effect of group, $F_{[1,17]} = 8.27$, $p = 0.01$, $\eta^2 = 0.32$), with no significant interaction ($F_{[2,34]} = 0.29$, $p = 0.75$, $\eta^2 = 0.017$).

To further investigate the underlying changes induced by the mPFC inactivation, we fitted the distributions of response durations with a double Gaussian function (*Figure 4D*). The distributions became bimodal during learning, and it has been shown (*Reyes et al., 2020*) that such a phenomenon happens not only at the group level—what could suggest an artifact of group averaging—but also at the individual level. Even well-trained animals display bimodal distributions, suggesting that rats alternate between responses of two classes: premature (short) and time-controlled (long, *Figure 4D*). We investigated how the two classes of responses evolve using the five parameters that the double Gaussian yields: $\gamma \in [0, 1]$, $\mu_1$, $\sigma_1$, $\mu_2$, and $\sigma_2$ ($\mu_1 \leq \mu_2$). The parameters μ and σ are the mean and the standard deviation of the Gaussian distributions, and γ represents the ratio between their amplitude. We adjusted a double Gaussian curve for each animal and session and plotted the extracted parameters as a function of the session.

The double Gaussian analysis confirmed that learning was impaired in the muscimol group and revealed that the effect was mainly on the temporally controlled responses (*Figure 4E–I*). Over sessions, although the temporal-controlled responses (mean of the second Gaussian, $\mu_2$) became longer for both groups, this increase was larger for the saline group (main effect of group, $F_{[1,17]} = 7.12$, $p = 0.016$, $\eta^2 = 0.30$; main effect of session: $F_{[2,34]} = 11.20$, $p = 0.020$, $\eta^2 = 0.40$; interaction: $F_{[2,34]} = 0.72$, $p = 0.49$, $\eta^2 = 0.04$). The standard deviation of the temporally controlled responses ($\sigma_2$) also evolved differently in the saline group (main effect of group, $F_{[1,17]} = 3.21$, $p = 0.09$, $\eta^2 = 0.16$; main effect of session: $F_{[2,34]} = 17.05$, $p < 0.001$, $\eta^2 = 0.50$; interaction: $F_{[2,34]} = 4.82$, $p = 0.04$, $\eta^2 = 0.22$).

The premature responses (associated with the first Gaussian) became shorter for both groups, that is, $\mu_1$ decreased (main effect of group, $F_{[1,17]} = 0.70$, $p = 0.41$, $\eta^2 = 0.04$; main effect of session: $F_{[2,34]} = 4.16$, $p = 0.024$, $\eta^2 = 0.19$; interaction: $F_{[2,34]} = 2.05$, $p = 0.14$, $\eta^2 = 0.11$). Neither $\sigma_1$ or γ

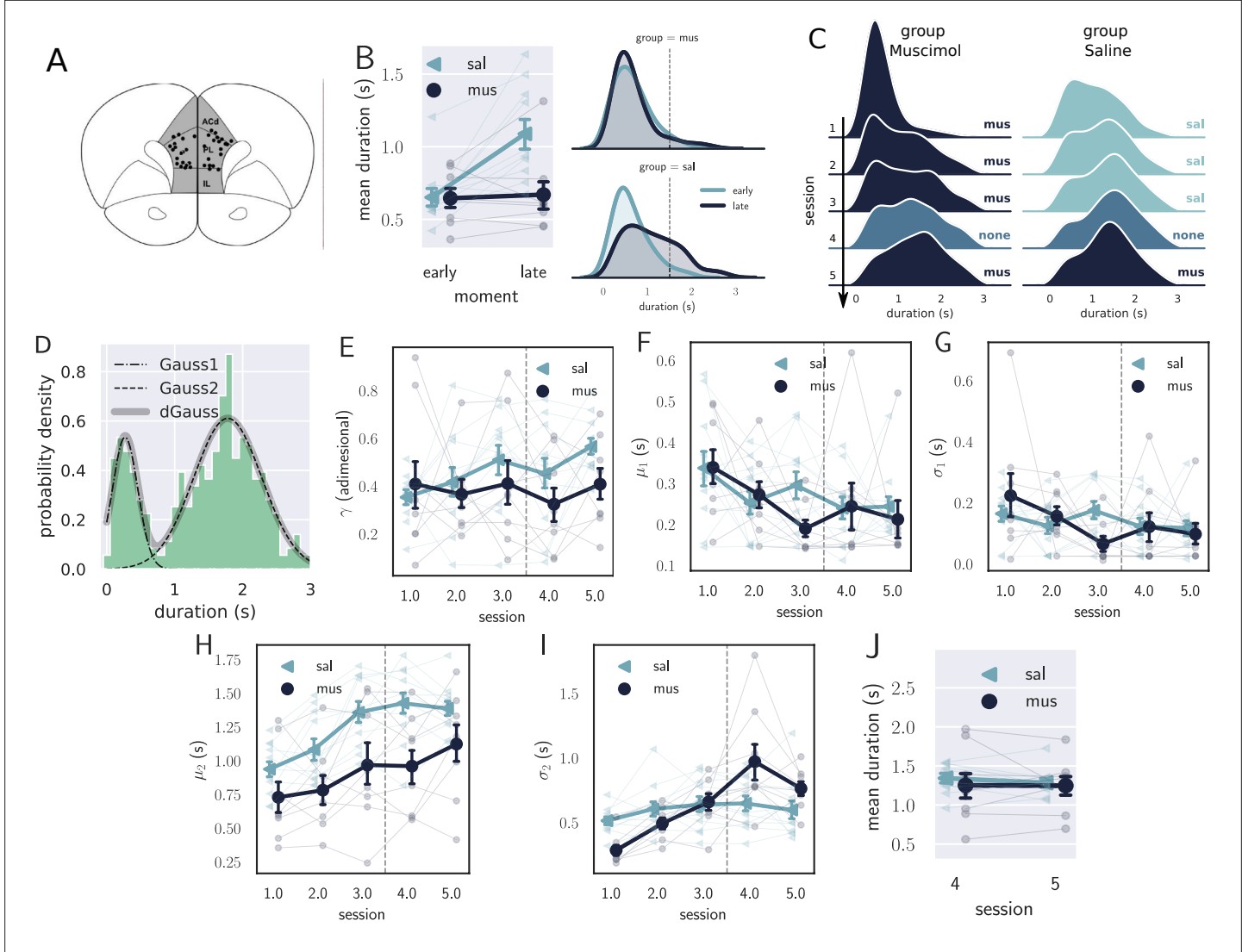

**Figure 4.** Results from the pharmacological inactivation of the medial prefrontal cortex (mPFC). (A) Histology showing the microinjection sites in the mPFC. (B) Left: average response duration early and late in the first session for saline (sal, N=8) and muscimol (mus, N=11) groups. Error bars represent the SEM. Right: same data as in left, but combining the responses from all rats for each group. The saline (sal) group increased the response duration during the first session (day 1), while the muscimol (mus) group produced no detectable change in behavior. (C) Probability density of late responses combined from all rats for the five consecutive sessions. The drug injected in each session is at the right of each graph and color-coded. The saline group evolves as expected, producing longer responses from the first session, producing bimodal distribution, whose second peak increases with respect to the first, remaining stable in sessions 3 and 4. The muscimol group has no sign of learning in the first session but progressed after the second session at a slower rate compared to saline group. By session 3, the peak of premature responses is still more prominent than the long (> 1 s) responses. (D) Histogram of response durations from one rat showing the bimodal distribution and the separate fitted distributions representing the premature responses (Gauss1), the temporally controlled responses (Gauss2), and the double Gaussian fit (dGauss). (E–I) Evolution of each double Gaussian parameter as a function of the sessions, γ (E), $\mu_1$ (F), $\sigma_1$ (G), $\mu_2$ (H), and $\sigma_2$ (I). The vertical dashed line divides the experiment into two phases: the first (sessions 1–3), when groups received different treatments, and second (sessions 4 and 5) when both groups received the same treatment. (J) Comparison between sessions 4 (no drug) and 5 (muscimol) for the saline (sal) and muscimol (mus) groups, suggesting that the muscimol injection produced no effect on trained animals.

were significantly modulated across the first three sessions ($\sigma_1$: main effect of group, $F_{[1,17]} = 0.05$, $p = 0.81$, $\eta^2 = 0.003$; main effect of session: $F_{[2,34]} = 1.45$, $p = 0.27$, $\eta^2 = 0.78$, interaction: $F_{[2,34]} = 2.76$, $p = 0.08$, $\eta^2 = 0.14$; γ: main effect of group, $F_{[1,17]} = 0.20$, $p = 0.66$, $\eta^2 = 0.01$; main effect of session: $F_{[2,34]} = 1.33$, $p = 0.28$, $\eta^2 = 0.07$, interaction: $F_{[2,34]} = 0.86$, $p = 0.42$, $\eta^2 = 0.04$). Overall, these differences point to decreased temporal control when the mPFC is inactivated.

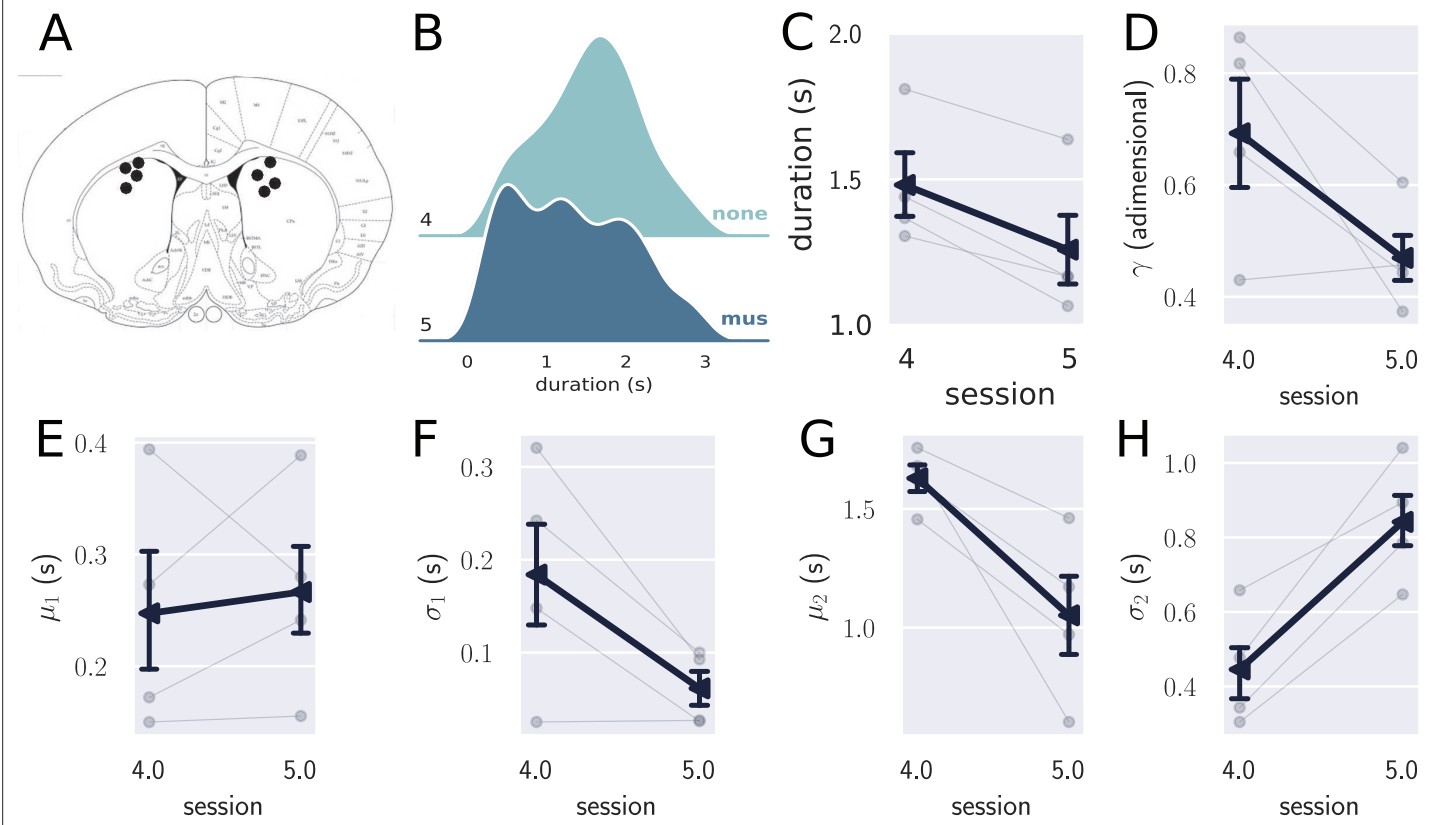

**Figure 5.** Inactivation of the striatum (STR) in trained animals. (**A**) Histological sites for the pharmacological experiments in the STR. (**B**) Comparison between the combined distribution of responses for trained rats (N=4) in session 4 (no drug) and session 5 (muscimol), showing that the temporally controlled responses were more variable when the STR was inactivated. (**C**) Mean response duration for sessions 4 (no drug) and 5 (muscimol). (**D–H**) Parameters obtained with the double Gaussian fit for sessions 4 and 5. The variables $\mu_2$ and $\sigma_2$ differed in these sessions, showing that the temporally controlled responses spread more with the inactivation of the STR. Errorbars represent the SEM.

The last phase of the experiment (sessions 4 and 5) aimed at investigating the effect of the mPFC inactivation on trained animals. By session 4—when rats received no injections—the rats from both groups were well trained, mostly producing temporally controlled responses. We compared the results from sessions 4 (no injection) and 5 (muscimol) to check how the inactivation of the mPFC affected the response distributions after learning.

The results from the last phase showed that inactivating the mPFC after training produced no detectable effect on the response distributions (***Figure 4J***). A mixed ANOVA yielded no significant effect for session ($F_{[1,17]} = 0.09$, $p = 0.77$, $\eta^2 = 0.005$), group ($F_{[1,17]} = 0.55$, $p = 0.47$, $\eta^2 = 0.03$), or interaction between the factors ($F_{[1,17]} = 0.033$, $p = 0.86$, $\eta^2 = 0.002$). Such a result indicates that the mPFC is only required at the beginning of the task—during the learning phase—becoming unnecessary as the animals get proficient. This finding is consistent with our previous observation regarding the disengagement of the mPFC observed in the electrophysiological recordings within the first session.

## Striatum is necessary for temporally controlled responses

Results from the decoding analysis also suggested a higher participation of the STR in the temporal task in the second training session. To confirm this finding, we inactivated the STR in proficient animals, after four sessions of training. STR inactivation strongly impaired temporal performance, leading to a lower mean response duration ($t(3) = 5.6$, $p = 0.011$, Cohen's d = 0.92, two-tailed $t$-test, ***Figure 5B and C***). We again fitted double Gaussians and compared their parameters across sessions 4 and 5 (before and during STR inactivation, respectively, ***Figure 5D–H***). We observed no significant effect for gamma ($t(3) = 2.3$, $p_\gamma = 0.11$), $\mu_1$ ($t(3) = -0.38$, $p_{\mu_1} = 0.72$), or $\sigma_1$ ($t(3) = 2.6$, $p_{\sigma_1} = 0.08$), but significant decrease in $\mu_2$ ($t(3) = 3.36$, $p_{\mu_2} = 0.04$), and an increase in $\sigma_2$ ($t(3) = -5.66$, $p_{\sigma_2} = 0.01$).

In sum, the inactivation of STR led to a decrease in $\mu_2$ to a level comparable to that observed in the very first session of training, while $\sigma_2$ increased to a level higher than that observed in session 1. Such changes suggest a significant decrease in the time precision of the temporally controlled responses. Interestingly, mean and standard deviation of the premature responses (first Gaussian) did not change in the inactivation session, which suggests that the STR was more involved with the production of time-controlled than in the premature responses.

## Discussion

In this study, we investigated mPFC's and STR's roles and activity while rats learned a temporal task. We tested the hypothesis that the time-related neuronal patterns previously reported in trained animals should emerge, especially in the mPFC, as rats learned to correctly time their responses. Differently from our hypothesis, results showed an initial involvement of the mPFC in the task (with a lower participation of the STR early on), but a diminishing role as the rats quickly became proficient in the task (with a higher participation of the STR after learning). These findings were consistent both from a decoding analysis of neural activity from mPFC and STR during learning, and from selective inactivation of those areas during learning.

The decoding analysis revealed a better performance early in the task for the mPFC compared to the STR neurons, but better decoding performance for the STR than mPFC cells later in the task. Correspondingly, pharmacological inactivation of mPFC showed that this structure was necessary for learning but not the expression of timed behavior, while the inactivation of STR after learning led to impairment of performance.

The decoder approach has been proposed to study the information encoding in different brain regions. *Bakhurin et al., 2017* showed that these algorithms could quantify the amount of information encoded in different mice brain regions—the STR and the orbitofrontal cortex—in a Pavlovian conditioning task. They concluded that both regions encoded time, but the STR outperformed the orbitofrontal cortex. We cannot compare our results directly with those from Bakhurin et al. because they recorded from well-trained animals (5–10 training sessions), from a different species, and registered in a different brain area (although still within the mPFC). However, on the second day of training, our results show that the STR outperforms the mPFC, a situation likely to remain stable since most learning happens on the first day of training (*Reyes et al., 2020*). Hence, it is possible that after the first day (or days) of training, the STR became even more reliable than the cortex in encoding the time of events, which agrees with results from *Bakhurin et al., 2017*.

Previous studies attempted to describe the collective reorganization of neuronal activity by recording in naive animals during their first day of learning (*Modi et al., 2014*; *Laubach et al., 2000*; *Komiyama et al., 2010*). *Modi et al., 2014* recorded from CA1 hippocampus neurons in a classical task—the trace eye-blink conditioning. They used calcium imaging techniques to identify clusters of neurons sequentially firing during the acquisition. They reported a progressive increase in firing sequences within the trial, concomitantly to improvements in behavioral performance. They also used the noise-correlation technique to infer common inputs to the network. Interestingly, the sequential firing observed in learners was preceded by a transient increase in noise correlations, which can be interpreted as a transitory increase in a common input to the CA1. Even though their behavioral task differs from ours, and the absence of monosynaptic projections from mPFC to CA1 (*Sesack et al., 1989*), these common transient effects seem an essential part of learning, and there might be a causal connection between them.

The decoding method proved to be useful in detecting changes in the network dynamics due to learning. Surprisingly, the neuronal activity in the mPFC was quite organized from the very beginning, shown by a good decoding performance early on the first day of training. Such performance may relate to the fact that animals practiced nose poking during the fixed ratio 1 (FR1) training phase. Hence, animals may have habituated to the action (nose poking), and the only new contingency introduced in the task was the wait time. In this sense, our results agree with previous results implicating the mPFC in top-down control of executive tasks, including behavioral switching learning (*Emmons et al., 2020*; *Antzoulatos and Miller, 2011*; *Freedman et al., 2001*; *Baker and Ragozzino, 2014*).

A confounding factor in our experiments is whether the results derive from learning temporal aspects of the task or from learning operational factors, such as attention and movement optimization. Even though electrophysiological results cannot isolate these effects, we suggest that our

training protocol favors the dissociation of temporal and nontemporal learning. First, animals were trained in FR1 sessions before initiating the Differential Reinforcement of Response Duration (DRRD) training phase. We argue that animals learn most of the operational aspects of the task during the FR1 procedure because nontemporal learning takes less time to train. For example, rats learn to press a bar and associate this activity with reinforcement in tens of trials, while they take hundreds of trials to produce timed responses. Second, in the DRRD procedure, the motor activity in each trial is highly time-locked with the trial, happening right before the trial begins ($t = 0$ ms) and right before the end of the trial. Finally, we removed 200 ms from the beginning and at least 200 ms at the end of each trial, aiming to exclude motor activity from the recordings.

Our pharmacological results provided causal evidence of the mPFC role in rats' ability to time longer responses. They revealed a substantial impairment in learning when the mPFC was inactivated, but this effect did not last after the first session. During the second session, rats from the muscimol group significantly improved during training, revealing that after the first session, rats learned—at least partially—even with their mPFC inactivated. One possible explanation is that the pharmacological procedure only partially inactivated the mPFC, and the remaining active neurons provided some learning. Also, other cortical areas may underlie the initial learning phase of this task, hence producing the observed change in behavior. Finally, the number of converging projections to the STR (*Shipp, 2017*) makes it prone to detect environmental changes from multiple brain sources beyond the mPFC. Hence, the STR may provide a different action selection and, consequently, a different strategy to optimize reward rate (*Antzoulatos and Miller, 2011*; *Oemisch, 2018*).

Another result of the inactivation of the mPFC was that this region completely disengages from timing after learning. Inactivation of mPFC did not disturb performance if the rats had already learned the task. *Narayanan et al., 2006* reported similar results in a different paradigm, a reaction time procedure. They showed that the inactivation of mPFC had an effect on the premature responses, but no effect on the time-controlled responses. Smith and collaborators also showed that the mPFC disengages from the task during habit formation (*Smith et al., 2010*). Our results also suggest that the mPFC may have a role similar to the observed in motor learning, where the cortex helps guiding plasticity downstream during initial phases, becoming unnecessary later on (*Kawai et al., 2015*). However, these results are at odds with several results showing the critical role of the mPFC in timing (*Kim et al., 2009*; *Buhusi et al., 2018*; *Dietrich and Allen, 1998*; *Xu et al., 2014*; *Kunimatsu and Tanaka, 2012*). Particularly, results from different tasks show that this region's inactivation impaired the timing precision but not accuracy (*Buhusi et al., 2018*; *Kim et al., 2009*). Then, since we dealt with a timing task, it seemed natural to assume the mPFC would be necessary for the expression of timed behavior. Such a hypothesis did not hold since inactivating the mPFC produced no detectable effect on experienced animals.

The mPFC's disengagement raises the question of why our results differ from those in the literature. One possible explanation has to do with the different time scale of our experiment. The 1.5 s interval used in our task is shorter than that used in most timing experiments. The interval of 1 s is usually referred to as the point of division between interval-timing and the millisecond range (*Buhusi and Meck, 2005*). Our interval is slightly above 1 s, and yet our results regarding the mPFC's role are quite distinct from experiments of more extended intervals (*Buhusi et al., 2018*; *Kim et al., 2009*). However, we cannot attribute the differences exclusively due to the interval duration. *Xu et al., 2014* used similar intervals to ours—training animals to reproduce auditory stimuli either 1.5 or 2.5 s long—and found that neuronal activity scaled with the estimated interval. They also manipulated the temperature of the mPFC, which biased the responses in time, providing a causal link between the mPFC and the time estimation. We hypothesize that the difference between results from Xu and colleagues and ours rests on the fact that our task is self-timed and self-initiated, that is, it does not rely or depend on external stimuli, which may give the task a more autonomic character, and reducing the number of required associative links to its execution.

Our results could not find evidence that the STR assumes the time encoding during the first session, even though rats learn the task and the mPFC disengages from it. It implies that other regions may be taking over such encoding. Results from *Heys et al., 2020* show that inactivating the medial entorhinal cortex (MEC) impairs timing learning. Even though they used longer intervals, MEC seems a candidate to mediate learning in our experiments. We also suggest that the thalamus may relate

to our results because of its implication in timing tasks and its neuronal adaptation under changes in temporal contingencies (*Komura et al., 2001*; *Lusk et al., 2020*; *Tanaka, 2007*).

The increasing involvement of the STR in the second day of training seen in our electrophysiological experiments was expected since the known role of this region in the expression of timed responses (*Mello et al., 2015*; *Matell et al., 2003*; *Monteiro, 2020*). However, our results give a timescale to this phenomenon, showing that the increase in time encoding happens already on the second training day. *Monteiro, 2020* showed a causal relationship between the dynamics within the STR and the decision based on timing in a temporal classification task, whose decision interval was 1.5 s. They manipulated the STR temperature showing an underestimation of time in increased temperatures and overestimation when the STR was cooled, consistent with the interpretation that the intrinsic dynamics of the STR underlies time estimation. When we inactivated the STR, the timing performance was also disrupted, providing another causal link between the STR function and timing. The STR inactivation affected both the accuracy and precision of the responses, strengthening the view that the STR has a central role for performance in the DRRD task after learning and deepening the understanding that the STR robustly underlies time estimation in various timing modalities.

Finally, our results agree with recent neuronal recordings during learning of a new interval in a fixed-interval task (*Emmons et al., 2020*). They showed that the mPFC and STR play different roles on temporal and behavior flexibility. While mPFC was more likely related to the behavior flexibility, STR was modulated by the temporal rule.

Overall, our results provide a double dissociation between the role of the mPFC and the STR in a timing task. They also further our understanding of regions involved in time estimation and production, improving our knowledge of a taxonomy of time (*Pöppel, 1978*; *Paton and Buonomano, 2018*).

## Methods

### Subjects

The subjects were 61 adult naive male Wistar rats 12–16 weeks old and weighing between 350 and 400 g (purchased from the Federal University of São Paulo, Brazil). In total, 18 animals were used in the STR pharmacological protocol, 35 for mPFC pharmacological protocol, and 8 in the electrophysiology protocol (four implanted in the mPFC and four in the mPFC and STR). Animals were housed individually in a 12 hr light/dark cycle (lights on at 7 am). All procedures were conducted during the light cycle. Rats were gradually food-deprived to reach and maintain 85% of their free-feeding weight. Water was freely available. All experimental protocols were approved by our institutional animal care and use committee (CEUA-UFABC).

### Apparatus

Animals with chronic implants of electrode arrays were trained in an operant chamber developed in our laboratory, controlled by an Arduino Uno board. The box was built of acrylic plastic. On one wall, a nose poke was equipped with an infrared emitter-sensor beam, which interrupts when the rat inserts its snout. There was a bottle with a metallic nozzle on the left of the same wall, from which the rat can lick to obtain the reward. This nozzle connects to a touch-sensitive electronic device that counts the number of licks. A metallic plate controlled by the Arduino circuit moves up and down, blocking or releasing the rat's access to the nozzle.

We used six MED-Associates operant chambers for the pharmacological procedure equipped with two levers, two cue lights (one above each lever), and a food cup on the front wall; more details are presented in *Reyes et al., 2020*. The rats executed an identical protocol as the rats trained in the electrophysiology boxes, except that the responses were produced on a lever and the rats were reinforced with a sugar pellet. The response duration was measured from the time the rats pressed the lever to the moment they released it.

We performed the experiments using different operant chambers due to resource optimization. Furthermore, the electrophysiological recordings were more affected by movement artifacts using lever presses and by electrical artifacts during chewing.

## Timing procedure

Animals implanted with electrodes were trained in our custom-made chambers to respond to the nose poke and receive access to three licks of 50% sucrose solution as a reward. Animals in the pharmacological experiments used the Med-Associates training chambers, responded on the left lever, and received a 45 mg sugar pellet as a reward. Regardless of the chamber used, the behavioral procedure was the same. First, the animals were auto-shaped to nose poke (press a lever) in an FR1 schedule of reinforcement, in which the animal received access to the glucose solution (sugar pellet) after each response. Rats received 60 min daily sessions in this schedule until they responded 100 times in a single session. In the next session, animals started the timing procedure described in *Reyes et al., 2020*. Trials were self-initiated and self-ended by responding on the nose poke (lever press). The animal had to nose poke (sustain the lever pressed) for at least 1.5 s to receive the reward: three licks in a 50% concentrated glucose solution (one 45 mg sugar pellet). The reward was only available after the animal self-ended the trial. Responses shorter than 1.5 s produced no consequences, and the rat could immediately start a new trial. Even if the animal did not consume the reward at the end of the trial, it could start a new trial. Rewards were available until the animal consumed them.

## Pharmacology

Bilateral cannulas were implanted in the STR of 18 rats. Three rats were removed from the experiments because they did not learn the FR1 after five sessions, and 11 were eliminated due to problems during drug injections or cannula placement. We also implanted bilateral cannulas on mPFC of 35 rats (male Wistar). From this group, 1 rat was removed because it did not learn the FR1 after five sessions, 15 were removed because of problems with drug injections or cannula placement. The stereotaxic coordinates were mPFC: AP = +3.2 mm, ML = ± 0.5 mm, DV = –3.3 mm; STR: AP = 0.4 mm; ML = ±3 mm; DV = 4.0 mm. After 1 week of recovery from the surgery, the animals were food-restricted, and the experiments began. Infusion of muscimol or saline at 100 ng/0.5 µl was made 10 min before the beginning of the behavioral session. Injecting needles were inserted into the guide cannula, and 0.5 µl of infusion fluid was delivered per site at 0.5 µl/min rate of infusion. After infusion was complete, the injector was held in place for 1 min to allow fluid diffusion.

For the STR manipulation, we used a single injection after learning (fifth session), and for mPFC manipulation, we separated the animals into two groups. The muscimol group received muscimol during the first three sessions, no infusion in the fourth session, and muscimol infusion again in the fifth session. In contrast, the saline group received saline infusion in the first three sessions, no injection in session 4, and muscimol infusion in session 5.

## Unit recordings and data analysis

Electrophysiological recordings came from two different sets of experiments, both using 32 single-wire array electrodes, 50 µm in diameter. In the first (n = 4, rats 7–10), the amplification system and microwire arrays from TDT (Tucker-Davis Technologies, USA) using all electrodes were implanted unilaterally around the mPFC, Stereotaxic coordinates: AP = 2.5 to 4.6 mm; ML = 0.1 to 1.5 mm; DV = 3.3 mm. Signals were digitized at 25 kHz and bandpass filtered between 1 and 5 kHz for spike detection. A threshold of 2 standard deviations of the signal was used to select spikes. The spike sorting for the first experiment consisted of two steps, first online (during recordings) and then offline with the Open Sort TDT software, which uses a principal component analysis to cluster the spike shapes. First, we used a k-means method to select three clusters. Then, the cluster was adjusted by visual inspection; noisy clusters were excluded. The second set of experiments (n = 4, rats 3–6) made use of the Open Ephys recording system (*Siegle et al., 2017*) with simultaneous recordings from mPFC and STR (AP = –0.3 to 2.0 mm; ML = 3.0 mm; DV = 4.0 mm). For the spike sorting, we used the kilosort package for semi-automatic spike sorting (*Pachitariu et al., 2016*), which uses template matching for the identification of spiking clusters and Phy (https://github.com/cortex-lab/phy; *Rossant, 2022*) for manual selection of proper waveform clusters. We used waveform shapes and autocorrelograms to identify putative single cells. Noisy clusters, with a significant percentage of inter-spike intervals falling below 2 ms, were excluded.

We assessed the stability of the spike waveform throughout the recordings by selecting the first and last waveforms in the session, keeping only neurons for which waveforms remained unaltered. For the first experiment, with longer sessions, we selected the first 30,000 and the last 30,000 spikes. For

the second experiment, we filtered the spikes during the trials and selected 3000 in both moments, beginning and end of the session. Then, we plotted the mean and standard deviation of these waveforms. The neuron was considered stable when the mean of the waveforms at the end of the session was at least inside the standard deviation shadow of the waveforms at the beginning of the session. Moreover, cells with abrupt changes in spike rate were removed. These two previous analyses detected seven unstable neurons, which were eliminated from the analysis.

## Surgical and histological procedures

Animals were anesthetized using intraperitoneal (IP) injections of a cocktail of ketamine (100 mg/kg) and xylazine (10 mg/kg). The surgical level of anesthesia was maintained using ketamine supplementary doses of 0.1 ml when the surgical duration was above 40 min or if the animal responded to stimulus on the tail or feet. While the animal was on a superficial anesthetic plan, we started a sepsis protocol and accommodated the animal on the stereotaxic apparatus. When the animal was under a deep anesthetic plan, we made an incision. Craniotomies were drilled above the mPFC or STR, from where the electrodes or cannulas were implanted. Also, four holes were drilled for skull screws, which were used to implant the ground wire to keep the implant's stability. After the electrode was placed into mPFC or STR, the craniotomy was closed with dental acrylic. Rats recovered for 1 week before the behavioral training.

Following the experiment, rats were anesthetized and euthanized with an injection of 100 mg/kg of urethane. To map the electrode position, we used an electrical current to promote damage where the electrodes were implanted. Moreover, to map the cannula position, the animal received a microinjection of methylene blue dye. Then we transcardially perfused the rats with 4% formalin. Brains were post-fixed in a solution of 8% formalin; after 2 days, the brains were transferred to a 30% sucrose solution until they sunk to the bottom of the Falcon tube. Next, they were frozen in isopentane at −80°C and sectioned in 25-μm slices in a cryostat. Brain slices were mounted on gelatin-subbed slides and stained for cell bodies using cresyl violet. Finally, the electrode and cannulae placements were defined by optical microscopy analysis.

## Data analysis
### Electrophysiological data
The neural activity analysis was performed in Python scripts developed in our lab. To calculate the peri-event raster and histograms, we computed the spiking activity from 0.5 s before the animals start (onset) a response (nose poke or lever press) until the moment they end the response (offset). Also, we calculated the peri-events for each neuron considering the neural activity aligned by the onset and offset of each trial. The peri-event histograms were calculated using bins of 100 ms and were smoothed with a bandwidth of 100ms.

### Climbing activity
We quantified the climbing activity in spikes with a linear fit on the firing rate of each neuron, as performed by *Kim et al., 2013*. The firing rate was computed using a Gaussian kernel density function for the selected trial vs. time binned in 100 ms. The fit was adjusted by the Python function 'linregress,' and we considered a neuron as a ramping neuron when the slope was significantly different from 0 with a 5% significance level.

### Multivariate pattern activity
We used separate multivariate analyses for the two groups of animals, one with recordings from mPFC and the other with simultaneous recordings from mPFC and STR. All analyses were performed on spiking data from all animals within groups, in a way that all registered cells were considered to be from a unique 'average' rat. We used the spikes from reinforced trials, that is, trials longer than the reinforcement criterion (1.5 s). Trials were truncated at the criterion time, and the average fire rate was calculated in bins of 100 ms.

When performing the spiking activity analysis with neurons from different rats, we first selected the rat with the smallest number of trials, let's say $N$. Then, we limited the trials from the other rats to $N$, so that we use the same number of trials for all rats. For the rat with the smallest number of trials, the

session was divided into two for selecting early and late trials (with $N/2$ trials, each). For the other rats, we designated the first $N/2$ trials as early, and the last $N/2$ as late trials (see *Figure 1B*).

We used a multiclass LDA as implemented in the sci-kit Python library, with least-squares solver, automatically calculated shrinkage using Ledoit–Wolf lemma, no priors, and the default number of components: min(number of classes – 1, number of features). Cross-validation was performed using 1000 random folds. In each fold, training data consisted of activity from 10 bins (one bin for each class) from 80% of the trials. Bins from the same trial were always in either the train or test set. Training and testing data were normalized based on the median and interquartile range for each neuron, calculated over the training set. The output for each fold was the probability of each class, which was in turn averaged across folds to generate the confusion matrices. Performance measures were estimated in each fold using a Pearson correlation from the true class with the highest probability class for each bin.

## Jackknife procedure

We also gauged the differences in classifier using a jackknife procedure inspired by previous work on ERP measurements (*Kiesel et al., 2008*). In sum, the differences in the classifier were calculated $N$ times, where $N$ is the number of animals in the group, removing one animal in each run. Then the $N$ different runs were compared with a *t*-test, obtaining the value of the T-statistics. The T-value is then divided by $N - 1$ (the jackknife correction), and this value was used to obtain the significance level of the comparison.

### Behavioral data

All data were analyzed using Python routines developed in our laboratory. Statistical analysis was performed in Jasp (*JASP Team, 2020*) and Pingouin (*Vallat, 2018*). Our dependent variable was response duration. For each animal, we constructed probability density diagrams of response duration. To test whether the animals learned the task in the first session, we compared a group of trials from the beginning and the end of the session. For the electrophysiology and STR protocol, we used paired *t*-tests. For the mPFC pharmacological protocol, we used a two-way ANOVA to compare groups and stages of the session.

As we showed in our previous work (*Reyes et al., 2020*), animals trained in the task frequently display a bimodal distribution, characterized by the persistence of very short responses interspersed with more timed responses—even after several sessions of training (*Platt et al., 1973*). To better characterize bimodal distributions, we used a double Gaussian fit. We fitted the responses of each animal and condition separately, using as ordinary least-squares method 'curve_fit' from the package 'scipy.optimize.' The initial parameters to start the fitting interactions were $\gamma = 0.5$, $\mu_1 = 0.2$, $\sigma_1 = 0.1$, $\mu_2 = 1$, and $\sigma_2 = 0.5$.

The double Gaussian probability density function was defined as $p(t_i) = f(t_i)/\xi$, where

$$f(t_i) = (1 - \gamma) \exp\left(-\frac{(t_i - \mu_1)^2}{2\sigma_1^2}\right) + \gamma \exp\left(-\frac{(t_i - \mu_2)^2}{2\sigma_2^2}\right), \tag{1}$$

and the normalization term

$$\xi = \Delta_t \sum_{i=1}^{N} f(t_i), \tag{2}$$

with time measured from 0 to 6 s in 0.1 s bins: $t_i = i * \Delta_t$, $i = 0 \dots 60$, $\Delta_t = 0.1$ s.

At the group level, estimated parameters were compared using parametric tests, such as *t*-tests and ANOVAs, indicated in each case.

## Acknowledgements

This work has been partially supported by grants #2016/18914-7, #2016/05473-2, #2018/20277-0, São Paulo Research Foundation (FAPESP), and grant #430993/2016-1, National Council for Scientific and Technological Development (CNPq). The authors thank Nandakumar Narayanan for valuable feedback on the manuscript.

# Additional information

## Funding

| Funder | Grant reference number | Author |
|---|---|---|
| Fundação de Amparo à Pesquisa do Estado de São Paulo | Graduate Student Fellowship 016/18914-7 | Gabriela C Tunes |
| Fundação de Amparo à Pesquisa do Estado de São Paulo | Graduate Student Fellowship 2016/05473-2 | Eliezyer Fermino de Oliveira |
| Fundação de Amparo à Pesquisa do Estado de São Paulo | 2018/20277-0 | Marcelo Bussotti Reyes |
| Fundação de Amparo à Pesquisa do Estado de São Paulo | 2017/25161-8 | André M Cravo |
| Conselho Nacional de Desenvolvimento Científico e Tecnológico | 430993/2016-1 | Marcelo Bussotti Reyes |
| Fundação de Amparo à Pesquisa do Estado de São Paulo | Grant #2014/50909-8 | Marcelo S Caetano |
| Conselho Nacional de Desenvolvimento Científico e Tecnológico | Grant No. 465686/2014-1 | Marcelo S Caetano |
| Fundação de Amparo à Pesquisa do Estado de São Paulo | Grant # 2017/03729-2 | Eliezyer Fermino de Oliveira |

The funders had no role in study design, data collection and interpretation, or the decision to submit the work for publication.

## Author contributions

Gabriela C Tunes, Conceptualization, Data curation, Formal analysis, Investigation, Visualization, Methodology, Writing – original draft, Writing – review and editing; Eliezyer Fermino de Oliveira, Conceptualization, Data curation, Formal analysis, Methodology, Writing – review and editing; Estevão UP Vieira, Software, Formal analysis, Validation, Investigation, Visualization, Methodology, Writing – review and editing; Marcelo S Caetano, Conceptualization, Methodology, Writing – review and editing; André M Cravo, Conceptualization, Software, Investigation, Visualization, Methodology, Writing – review and editing; Marcelo Bussotti Reyes, Conceptualization, Resources, Data curation, Software, Formal analysis, Supervision, Funding acquisition, Validation, Investigation, Visualization, Methodology, Writing – original draft, Project administration, Writing – review and editing

## Author ORCIDs

Gabriela C Tunes ![ORCID] http://orcid.org/0000-0002-8701-4258
Eliezyer Fermino de Oliveira ![ORCID] http://orcid.org/0000-0002-9651-8570
Estevão UP Vieira ![ORCID] http://orcid.org/0000-0003-1232-0805
Marcelo S Caetano ![ORCID] http://orcid.org/0000-0002-5353-7982
André M Cravo ![ORCID] http://orcid.org/0000-0002-8580-5697
Marcelo Bussotti Reyes ![ORCID] http://orcid.org/0000-0001-7811-7696

## Ethics

All of the animals were handled according to approved institutional animal care and use committee (CEUA) of the Universidade Federal do ABC. This same committee approved the experimental protocols (Permit Numbers: 001/2014 and 020/2015). We made all the efforts to minimize animal suffering.

## Decision letter and Author response

Decision letter https://doi.org/10.7554/eLife.65495.sa1
Author response https://doi.org/10.7554/eLife.65495.sa2

## Additional files

### Supplementary files

• Transparent reporting form

• Supplementary file 1. Classification analysis on incorrect trials. The analysis shown here was identical to that in *Figure 3* of the main manuscript, except that we only used trials that were shorter than 1.5 s (non-reinforced) and longer than 1s.

### Data availability

All behavioral and spiking-time data have been deposited in https://osf.io/uh4s6/ and are publicly available.

The following previously published dataset was used:

| Author(s) | Year | Dataset title | Dataset URL | Database and Identifier |
|-----------|------|---------------|-------------|-------------------------|
| Reyes M | 2020 | Code Migration | https://osf.io/uh4s6/ | Open Science Framework, uh4s6 |

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
