## [Editor Report]

This study investigates the question of whether distinct brain areas differentially encode time during the learning of a simple motor timing task. The key novel result is that early in training the dynamics of the medial prefrontal cortex provides the best code for time, but later in training the striatum provides a better code. In addition, the article reports that the inactivation of medial prefrontal cortex produces a delayed learning effect, while the inactivation of the striatum after learning led to impairment of performance. Thus, the observation that temporal coding and the necessity of brain area for task performance transfers from medial prefrontal cortex to the striatum during learning is an important observation for the field.

---

## [Decision Letter]

**Decision letter after peer review:**

Thank you for submitting your article "Time-Encoding Migrates from Prefrontal Cortex to Dorsal Striatum During Learning of a Self-Timed Response Duration Task" for consideration by *eLife*. Your article has been reviewed by 3 peer reviewers, including Hugo Merchant as the Reviewing Editor and Reviewer #1, and the evaluation has been overseen Richard Ivry as the Senior Editor.

We have also prepared an Evaluation Summary and Public Reviews of your work below, which are designed to transform your manuscript into a preprint with peer reviews.

Essential revisions:

We have the following recommendations about the data set and methods in order to critically evaluate the robustness of their results.

1) Decoding analysis.

a. Please report the number of units recorded in each animal/session. All statistics were performed on data resulting from a decoder applied to neural data, and thus the degrees of freedom reflected in the reported F statistics of their ANOVAs would appear to correspond to folds in a cross validation procedure. It would be important to know more precisely how the differences in decodability of certain variables relates to the number of units recorded.

b. All analyses of neural data are performed on data pooled across animals. This makes it difficult to determine whether the effects they observe are consistent across animals. The authors are attempting to analyze data from single sessions, and thus they may have small amounts of data from single animals. In the present form it is difficult to critically evaluate the consistency and robustness of their observations. Within-animal analyses would go a long way towards resolving this issue.

c. In Figure 3 the statistical analyses shows highly significant group effects and an interaction (F(1,99) = 374). But the stats seem to be done on a per trial basis? If this is the case it is not clear to me if this is correct, as opposed to relying on the mean correlation across trials for each animal? Perhaps the authors did it this way because they collapsed neurons across animals? Either way it is necessary to clarify theses analyses and perhaps perform additional analyses depending on the answer to the questions above.

d. Given that the authors are highlighting changes in decodability within a session, it is important to assess that recording quality was constant over the session, for example determining whether they observed non-stationarity in firing rates during the sessions and/or changes in spike waveform shape. Ideally this would be applied to baseline activity outside of a trial. Indeed, more information about steps taken to guarantee good unit isolation would be useful.

e. It seems that both areas encode the beginning and end of the trials, with high densities in the diagonal only on the initial and final bins (Figure 3B and E), rather than the elapsed time across all the trials. These results could be related with learning of non-temporal factors discussed below.

f. The decoding of elapsed time both areas went down from early to late trials in the experiment of one session (Figure 3C and D), supporting the notion that the striatum does not take over, although the rats learned to time the interval (Figure 1B and C). Which potential brain areas are involved in this short learning process then?

2) Learning Process. It is difficult to dissociate the role of mPFC and the striatum linked with a better representation of elapsed time with learning from the operational learning aspects of the task. The latter include the increase in attention of sensory inputs associated with the nosepoke, an increase in precision of movement kinematics (less body and face movements during the nose poke), and a more developed reward expectation from learning to time the 1.5 s. The authors should perform careful analysis to try to dissociate the learning of temporal and non-temporal factors and the involvement of the two areas.

Recommendations for the authors.

1) How did the authors define "early" and "late" periods of sessions? I may have missed it, but I could not find this information in the paper. I assume also that "early" and "late" correspond to the "moment" factor that they include in their statistical tests. Relatedly, it would be useful to define clearly in Figure 1B the division between early and late trials.

2) It was not clear how the climbing activity was quantified, and what the N values on lines 63-63 mean (and why the units seem to be in seconds?).

3) The striatum shows an increase in decoding on the second day experiment, is this an effect of the total number of trials executed by the animals? Which brain areas could be linked to the one day learning of 1.5 timing then?

4) The unimodal distributions in Figure 1B are replaced by bimodal distributions in Figure 4C. Is this an effect of changing the effector from nosepoke to lever press?

5) Figure 2 rasters are fine, but PSTHs seem to be a bit misleading… PSTH heights drop towards the center of the plots because there are fewer and fewer trials with data in those bins. Avg sp/s should be normalised based on the number of trials with data in each bin.

6) What is the relation between early decoding of session 2 and the late decoding of session 1? In the behavior there is a clear carry over of learning (Figure 1C).

7) The split positive/negative time axes in Figure 2C-J need to be explained better.

8) Please report the posterior probabilities of decoded times, are they above chance level?

Is the decoding more accurate with SVMs than the used Linear Discriminant Analysis?

9) Please perform the decoding on incorrect trials below 1.5 seconds. Are the results different from those reported in Figure 3?

10) Figures 3C,D and F should have the same scale.

11) Please state why the physiology and pharmacology experiments were performed in different behavioral boxes, employing nose port withdrawal or lever press as an operant response, respectively.

12) Mu2 and Sigma2 can be the behavioral fingerprints for time accuracy and precision. Is peculiar the animals become more accurate on timing 1.5s but not more precise with training. Please discuss.

13) All the literature cited on timing neurophysiology is on the rodent. Some references on non-human primates should be included.

14) The general observation that task-dependence shifts from cortex to striatum over learning would seem to be consistent with a series of studies from the laboratory of Bence Olveczky starting with Kawai et al. Neuron, 2015. Though they focus more on motor cortex, these studies should probably be cited.

15) Overall, the methods would benefit from a careful screen through the manuscript to make sure that any approaches and terms used in the paper are clearly defined in the methods.

16) The authors are using "moment" to refer to the early and late stages of training within a session. This is a bit confusing; I might recommend just using "stage".

17) The first sentences of the Discussion are a bit confusing … "previously reported" will make it clearer.

---

## [Author Response]

Essential revisions:We have the following recommendations about the data set and methods in order to critically evaluate the robustness of their results.1) Decoding analysis.a. Please report the number of units recorded in each animal/session. All statistics were performed on data resulting from a decoder applied to neural data, and thus the degrees of freedom reflected in the reported F statistics of their ANOVAs would appear to correspond to folds in a cross validation procedure. It would be important to know more precisely how the differences in decodability of certain variables relates to the number of units recorded.

We added to the text the number of units recorded (N=254), and the number of units tha specified by region either in the introduction and in the methods section (as a table).

Regarding the F statistics, please see the next two answers.

b. All analyses of neural data are performed on data pooled across animals. This makes it difficult to determine whether the effects they observe are consistent across animals. The authors are attempting to analyze data from single sessions, and thus they may have small amounts of data from single animals. In the present form it is difficult to critically evaluate the consistency and robustness of their observations. Within-animal analyses would go a long way towards resolving this issue.

As the reviewer pointed out, we are trying to decode intervals within a single session, which makes it difficult to have enough data from single animals. The decodability varies strongly with the number of neurons recorded (see simulations below), and we needed to pool neurons from different animals to accomplish the analysis.

However, we agree this is an important question, so we investigated a little further the effect of individual rats using a Jackknife procedure as we delineate below.

c. In Figure 3 the statistical analyses shows highly significant group effects and an interaction (F(1,99) = 374). But the stats seem to be done on a per trial basis? If this is the case it is not clear to me if this is correct, as opposed to relying on the mean correlation across trials for each animal? Perhaps the authors did it this way because they collapsed neurons across animals? Either way it is necessary to clarify theses analyses and perhaps perform additional analyses depending on the answer to the questions above.

We agree with the criticism regarding the degrees of freedom in the statistics. Hence, we removed the F statistics that used the folds in cross validation as degrees of freedom. We could not find a reference method to estimate the variability (between trials) of the decoder for the case where we incorporate all neurons and trials to the analysis. Hence we used the distribution of the 1000 (instead of the 100 in the first submission) bootstrap samples and its 95% confidence interval of the bootstrap distribution. This seems to be a good estimator if the difference between stages (early versus late or day1 versus day2) are not only due to trial variability.

We further evaluated the between-animal variability with a jackknife analysis, evaluating the two regions separately (PFC and STR). We removed each one of the N animas at a time from the analysis and calculated the decoding performance (Pearson R) in each condition (early and late in single sessions, day 1 and day 2 for two sessions). Then, we calculated the T-statistics from the N combinations and corrected the T-value according to the jackknife procedure (following the method shown in Kiesel et al., 2008). Removing one animal at a time avoids analyzing the data with too few neurons, providing a more robust analysis.

The only (marginally) significant result when analyzing animals separately was the decoding performance in the first session, when we use all 8 animals for the analysis. Yet, we believe that single neurons, and the analysis of neurons pooled from different animals provide evidence that support the changes in the decodability in PFC and STR. Such evidence was complemented with the pharmacological experiments, reinforcing the robustness of the evidence.

Reference: Kiesel, A., Miller, J., Jolicœur, P. & Brisson, B. Measurement of erp latency differences: comparison of single-participant and jackknife-based scoring methods. Psychophysiol. 45,250–274 (2008).

d. Given that the authors are highlighting changes in decodability within a session, it is important to assess that recording quality was constant over the session, for example determining whether they observed non-stationarity in firing rates during the sessions and/or changes in spike waveform shape. Ideally this would be applied to baseline activity outside of a trial. Indeed, more information about steps taken to guarantee good unit isolation would be useful.

We included the following description of the steps we took to check the stability in the neuronal activity in the methods session:

“We assessed the stability of the spike waveform throughout the recordings by selecting the first and last waveforms in the session, keeping only neurons for which waveforms remained unaltered. For the first experiment, with longer sessions, we selected the first 30,000 and the last 30,000 spikes. For the second experiment, we filtered the spikes during the trials and selected 3,000 in both moments, beginning and end of the session. Then, we plotted the mean and standard deviation of these waveforms. The neuron was considered stable when the mean of the waveforms at the end of the session was at least inside the standard deviation shadow of the waveforms at the beginning of the session. Moreover, cells that had abruptly changed the overall mean of spikes in a trial at the beginning and the end, were removed. These two previous analyses detected seven unstable neurons, which were eliminated from the analysis. Below we show a sample of the waveform for one cell that met our criteria (and kept in our analysis) and one that was removed.”

In Author response image 1 we also show two examples of cells, one that was kept for the analysis, and the other that were excluded.

**Author response image 1. sa2fig1:** Figure with spikes from a neuron that remained stable throughout the session and met the criterion (left) and one that did not (right). In both figures, the average spike waveform from the beginning of the session is shown in blue and from the end in green. The corresponding shaded areas show the standard deviation of the waveforms.

e. It seems that both areas encode the beginning and end of the trials, with high densities in the diagonal only on the initial and final bins (Figure 3B and E), rather than the elapsed time across all the trials. These results could be related with learning of non-temporal factors discussed below.

Here we provide evidence that the fact that the bins at the beginning and the end of the trials are brighter (higher decoding value at extreme bins of the diagonal) does not relate to any particular non-temporal aspect of the spike trains. It rather reveals a particularity of the LDA classifier applied to this kind of data. The first evidence was that, no matter where we cut the trials after the nose poke onset (beginning) and before the end of the trial, the same effect occurs (results not shown). In the manuscript, we cut the trials at 0.2 s and 1.2 s and we used only the middle 1 s of the trials in the classifier (corresponding to the 10 bins shown in heatmaps). In previous versions of our analysis, we used the entire range of recordings (from -0.5 to 1.5 s), and we observed the same effect. We suspected that non-temporal effects might have been taking place and decided to drop the spikes close to the movement from the analysis. However, when we removed 200 ms from the beginning and the end of trials, as shown in the manuscript, the effect remained, strongly suggesting that the effect resulted from the method itself.

To double check these questions, we tested the method in a more controlled situation. We simulated the activity of ramping cells (used as a proxy of the encoding neurons in this simulation, see raster plat in Author response image 2) with spiking rates similar to the recorded neurons and applied the same decoder to the data.

**Author response image 2. sa2fig2:** Raster plot of a simulated ramping cell.

Applying the classifier to data consisting only of ramping neurons, the effect persists, even though, by construction, there is no particular activity happening at the beginning and at the end of each trial. See Author response image 3.

**Author response image 3. sa2fig3:** Figure showing the classifier applied to simulated data consisting of several ramping neurons and several trials. Each graph from A through E shows a result from a different number ofneurons (Neus.) and the number of trials in the simulated data. Generally speaking, the effect of higher decoding probability at the beginning and at the end of the trials persists. On the right of each heat map there is a histogram showing the correlation coefficient obtained in each fold of the data.

To further investigate the questions, we used the classifier data consisting of noise, i.e. spikes occurring randomly throughout the trials (see raster plot in Author response image 4). This would allow us both check possible biases of the classifier and if the previously observed effect could be related to ramping neurons. As we can see from the heat maps in Author response image 5, the several bins at the border of the classifier present elevated classification, strengthening the evidence that the effect happens in bins at the border.

**Author response image 4. sa2fig4:** Raster plot of a simulated neuron with uniform response probability density.

**Author response image 5. sa2fig5:** Heatmaps and histograms showing four different runs of the analysis, each for a different set of simulated data. Neurons were simulated as random spikes uniformly distributed over the entire trial duration. We still can see that many bins at the border (top or down) are brighter. However, the overall mean value of the r coefficient remains zero on average for all cases.

f. The decoding of elapsed time both areas went down from early to late trials in the experiment of one session (Figure 3C and D), supporting the notion that the striatum does not take over, although the rats learned to time the interval (Figure 1B and C). Which potential brain areas are involved in this short learning process then?

The reviewer brings an interesting question, which we gladly included in the manuscript (a paragraph in the discussion).

The literature on timing learning is scarce, particularly in naive animals. Our best guess relies on the study from Heys and collaborators, showing that inactivating the medial entorhinal cortex (MEC) impairs learning of an active interval timing task. Even though the tasks they used comprised longer intervals, MEC seems a good candidate. Our second guess is the thalamus, which has been implicated in timing tasks and whose behavior adapts under changes in temporal contingencies (Komura, et al. 2001, Lusk, Meck and Yin).

James G. Heys, Zihan Wu, Anna Letizia Allegra Mascaro, Daniel A. Dombeck, Inactivation of the Medial Entorhinal Cortex Selectively Disrupts Learning of Interval Timing, Cell Reports, Volume 32, Issue 12, 2020.

Komura, Yutaka, Ryoi Tamura, Teruko Uwano, Hisao Nishijo, Kimitaka Kaga, and Taketoshi Ono. “Retrospective and Prospective Coding for Predicted Reward in the Sensory Thalamus.” *Nature* 412, no. 6846 (August 2, 2001): 546–49.

https://doi.org/10.1038/35087595.

Lusk, N., Meck, W. H. & Yin, H. H. Mediodorsal thalamus contributes to the timing of instrumentalactions. J. Neurosci. 40, 6379–6388 (2020).

2) Learning Process. It is difficult to dissociate the role of mPFC and the striatum linked with a better representation of elapsed time with learning from the operational learning aspects of the task. The latter include the increase in attention of sensory inputs associated with the nosepoke, an increase in precision of movement kinematics (less body and face movements during the nose poke), and a more developed reward expectation from learning to time the 1.5 s. The authors should perform careful analysis to try to dissociate the learning of temporal and non-temporal factors and the involvement of the two areas.

We agree that there might be other (operational) processes taking place. However, the precautions we took were the following:

1) The animals passed through fixed ratio 1 (FR1) sessions before initiating the DRRD task. With that, we can argue that the operational aspects of the task took place mostly in the FR1 tasks. Furthermore, it seems natural to assume that the operational learning is less cognitively intense than the timing task, and hence animals tend to learn them faster.

2) In the DRRD task the motor activity in each trial is highly stereotyped. We placed the infra-red beam (that detects the beginning of the trial) at the end of the nose poke orifice. This forces that the trial starts when the animal is very close to the final position in the nose poke and the trial ends as soon as the animal starts moving out the orifice. Finally, we removed 200 ms from the beginning and more than 200 ms at the end and the end of each trial, exactly to reduce chances of “contamination” of the motor aspects on the neural activity.

We included a paragraph with this line of reasoning in the discussion.

Recommendations for the authors.1) How did the authors define "early" and "late" periods of sessions? I may have missed it, but I could not find this information in the paper. I assume also that "early" and "late" correspond to the "moment" factor that they include in their statistical tests. Relatedly, it would be useful to define clearly in Figure 1B the division between early and late trials.

We included a description in the methods session describing how we define early and late. We also updated Figure 1 as suggested. The included paragraph reads:

“When performing the spiking activity analysis with neurons from different rats, we first selected the rat with the smallest number of trials, let's say N. Then, we limited the trials from the other rats to N, so that we use the same number of trials for all rats. For the rat with the smallest number of trials, the session was divided in half for selecting early and late trials (with N/2 trials, each). For the other rats, we designated the first N/2 trials as early, and the last N/2 as late trials (see Figure 1B).”

(2) It was not clear how the climbing activity was quantified, and what the N values on lines 63-63 mean (and why the units seem to be in seconds?).

We included a new subsection in the methodology section to describe it in more detail:

“We quantified the climbing activity in spikes with a linear fit on the firing rate of each neuron, as performed by Kim and collaborators (Kim et al., 2013). The firing rate was computed using a gaussian kernel density function for the selected trial vs time binned in 100 ms. The fit was adjusted by the python function "scipy.stats.linregress", and we considered a neuron as a ramping neuron when the slope was significantly different from 0 with a 5% significance level.”

3) The striatum shows an increase in decoding on the second day experiment, is this an effect of the total number of trials executed by the animals? Which brain areas could be linked to the one day learning of 1.5 timing then?

We compared the day 1 and day 2 data only for the four rats that had implants in the STR, rats 3 to 6. They executed 582 (145 SEM) trials on average on day 1 and 536 (87 SEM) trials on day 2 (see Author response table 1). Hence the number of trials were quite similar and is probably not the cause of the difference.

**Author response table 1. sa2table1:** Table with the number of trials for each rat implanted in the STR.

Dayrat	1	2
3	543	625
4	341	381
5	509	437
6	936	699

Regarding the brain areas, as we suggested above, our hypotheses are the medial entorhinal cortex and the thalamus.

4) The unimodal distributions in Figure 1B are replaced by bimodal distributions in Figure 4C. Is this an effect of changing the effector from nosepoke to lever press?

The changing of the effector only partially impacts the learning of the animals. However, we believe that this change occurs because, in figure 1B, each distribution represents a moment of the session, while figure 4C represents the distribution of the whole session. Author response image 6 is a sample considering the whole session of the rat 6 using nosepoke, where one can see the bimodal distribution, which is typical for all trained rats, regardless of the effector.

**Author response image 6. sa2fig6:** 

5) Figure 2 rasters are fine, but PSTHs seem to be a bit misleading… PSTH heights drop towards the center of the plots because there are fewer and fewer trials with data in those bins. Avg sp/s should be normalised based on the number of trials with data in each bin.

The reviewer is correct. We reformulated the figure to calculate the firing rates based only in the active trials for each time bin.

6) What is the relation between early decoding of session 2 and the late decoding of session 1? In the behavior there is a clear carry over of learning (Figure 1C).

Our data provide some evidence that between the end of session 1 and the beginning of session 2, there is something like a discontinuity, i.e. the decoding performance in mPFC decays (even more than during session 1) and the performance in the STR increases. Hence, the decoding performance inverts from the end of one session to the beginning of the next, remaining unaltered during the second session. We believe that such discontinuity may be a result of “offline” learning, such as memory consolidation during sleep, for example. That would also explain why rats carry over a substantial learning, revealed by a good behavioral performance at the beginning of session two, as pointed out by the reviewer.

Author response image 7 shows the decoding performance on the two training days, broken down to early and late stages. We can see that there is an inversion between the decoder in the mPFC and STR.

**Author response image 7. sa2fig7:** 

7) The split positive/negative time axes in Figure 2C-J need to be explained better.

We added some sentences in the legend to improve the understanding of the figure.

8) Please report the posterior probabilities of decoded times, are they above chance level?Is the decoding more accurate with SVMs than the used Linear Discriminant Analysis?

We have previously tested several algorithms, including the SVM, as shown in Author response image 8. Even though LDA did not present the best decoding performance among the algorithms, it presented a good balance between simplicity and performance, and hence we chose to report only its results.

**Author response image 8. sa2fig8:** Figure comparing different algorithms decoding the time intervals, as the final pearson’s R coefficient. Methods used, from top to bottom: QDA: Quadratic Discriminant Analysis,Naive_bayes, rbfSVM:Radial Basis Function kernel SVM, Decision Tree, linSVM: linear SVM, Neural network, Random Forest, Logistic Regression, LDA: linear discriminant analysis, LightGBM: Light Gradient Boosting Machine, and XGboost.

9) Please perform the decoding on incorrect trials below 1.5 seconds. Are the results different from those reported in Figure 3?

We performed the classification analysis for trials that were longer than 1 s and shorter than 1.5 s. Then we trimmed the trials to analyze only the “central part of the trial” eliminating the motor preparation at the beginning and at the end of each trial, i.e. we analyzed the time from 0.2 to 0.8 s in bins of 0.1 s. As expected due to a reduced number of trials, the classification performance decreases, but the general tendency remains unaltered. We included the figure in the supplementary material (supplementary Figure).

10) Figures 3C,D and F should have the same scale.

We updated the figure so that all scales match.

11) Please state why the physiology and pharmacology experiments were performed in different behavioral boxes, employing nose port withdrawal or lever press as an operant response, respectively.

The difference in the experiments resulted from resource optimization. We had 6 boxes available for lever training and testing (with solid reward), but just one for nose poking and liquid reward. And, from our pilot experiments, we observed that lever pressing generates more movement noise, and solid reward produces electrical noise from muscle activation during chewing.

We added a statement in the methods section as requested.

Just as a comment, we had previously trained dozens of animals with levers (as a result of other experiments), providing us a good template for learning dynamics in this task. When we switched to the nose pokes, our pilot experiments showed quite similar results, including in the timescale of learning.

12) Mu2 and Sigma2 can be the behavioral fingerprints for time accuracy and precision. Is peculiar the animals become more accurate on timing 1.5s but not more precise with training. Please discuss.

The reviewer raises an interesting question. We believe this effect may occur because two opposing events are taking place. One is that the animals become more accurate with learning, which implies an increase in mu2. We also expect sigma2 to decrease with training, due to improvements in accuracy. As mu2 increases, sigma2 tends to increase as a consequence of the Weber law (aka scalar property of time). These two trends may cancel each other, resulting in a sigma1 constant as a function of the sessions, at least for these early stages of training.

13) All the literature cited on timing neurophysiology is on the rodent. Some references on non-human primates should be included.

We appreciate the comment and included more references as requested.

14) The general observation that task-dependence shifts from cortex to striatum over learning would seem to be consistent with a series of studies from the laboratory of Bence Olveczky starting with Kawai et al. Neuron, 2015. Though they focus more on motor cortex, these studies should probably be cited.

We agree that this reference fits in the manuscript and included it in the discussion.

15) Overall, the methods would benefit from a careful screen through the manuscript to make sure that any approaches and terms used in the paper are clearly defined in the methods.

We tried to improve the suggested points and some others. But we are open to improving any other points in case it is still not enough.

16) The authors are using "moment" to refer to the early and late stages of training within a session. This is a bit confusing; I might recommend just using "stage".

We modified throughout the text and in the figures.

17) The first sentences of the Discussion are a bit confusing … "previously reported" will make it clearer.

We modified the text according to the suggestion.